# Imperfect language learning reduces morphological overspecification: Experimental evidence

**Aleksandrs Berdicevskis[1¤]\*, Arturs Semenuks[2]**

**1** Department of Language and Linguistics, UiT The Arctic University of Norway, Langnes, Tromsø, Norway, **2** Department of Cognitive Science, University of California, San Diego, La Jolla, CA, United States of America

¤ Current address: Department of Swedish, University of Gothenburg, Gothenburg, Sweden
\* aleksandrs.berdicevskis@gu.se

**Data Availability Statement:** All relevant data are within the paper and its Supporting information files.

**Funding:** AB: Norwegian Research Council grant "Birds and Beasts" (222506), https://www.

## Abstract

It is often claimed that languages with more non-native speakers tend to become morphologically simpler, presumably because non-native speakers learn the language imperfectly. A growing number of studies support this claim, but there is a dearth of experiments that evaluate it and the suggested explanatory mechanisms. We performed a large-scale experiment which directly tested whether imperfect language learning simplifies linguistic structure and whether this effect is amplified by iterated learning. Members of 45 transmission chains, each consisting of 10 one-person generations, learned artificial mini-languages and transmitted them to the next generation. Manipulating the learning time showed that when transmission chains contained generations of imperfect learners, the decrease in morphological complexity was more pronounced than when the chains did not contain imperfect learners. The decrease was partial (complexity did not get fully eliminated) and gradual (caused by the accumulation of small simplifying changes). Simplification primarily affected double agent-marking, which is more redundant, arguably more difficult to learn and less salient than other features. The results were not affected by the number of the imperfect-learner generations in the transmission chains. Thus, we provide strong experimental evidence in support of the hypothesis that iterated imperfect learning leads to language simplification.

## 1. Introduction

### 1.1. Social structure and linguistic complexity

Linguistic diversity represents one of the biggest challenges for cognitive science [1]. Which cognitive biases and constraints (if any) and which social factors (if any) shape structural patterns that can be observed in human languages across the world? In general, how do extra-linguistic factors shape linguistic structure [2, 3] and can they shape linguistic structure at all? While some scholars embrace this possibility [4, 5], others insist that language cannot be

forskningsradet.no/en/. The funders had no role in study design, data collection and analysis, decision to publish, or preparation of the manuscript.

**Competing interests:** The authors have declared that no competing interests exist.

understood other than on its own terms [6]. Linguistic complexity is a convenient parameter with which to test theories about the role of extra-linguistic factors.

Recent years have witnessed increased interest in sociocognitive determinants of linguistic complexity [7–10]. As regards research questions, one of the goals is to explain the distribution of complexity across the languages of the world [11]. As regards linguistic domain, complexity within morphology is the most widely studied topic (see [12, 13] for a discussion of the reasons).

Several influential theories have emerged [11, 14–17] that link the likelihood of a given language to accrue or to lose morphological complexity to a range of social factors. The theories show a remarkable unity with regards to the key idea, which, while worded differently in different studies, can be reduced to the distinction between "normal" and "interrupted" language transmission (predominantly intergenerational transmission). Trudgill [11] lists five major social factors that can be viewed as interrupting or inhibiting transmission: large population size, high levels of contact (of the type involving adult bilingualism), loose social networks, low social stability and small amounts of shared information. These factors are likely to favour simplification, while their opposites (small population size, low levels of contact etc.) are not. In other words, in the case of interrupted transmission, complexity is likely to decrease, whereas in the case of normal transmission it is likely to stay constant or increase. In this study, we investigate whether imperfect learning following interrupted transmission, which could be caused by the factors listed above, leads to simplification (see more in Section 1.2).

The theories cited above rely mostly on typological evidence. Initially, the evidence came from qualitative generalizations made by typologists and sociolinguists on the basis of their observations [11, 12, 14, 16]. Later, a number of more rigorous quantitative studies followed which examined the correlation between various facets of complexity and social parameters, such as population size [18–20], share of non-native speakers [21, 22], both of these factors [23, 24], contact intensity [25], the geographical area, linguistic diversity and contact intensity within it [26], and the type of language: creole vs. non-creole [27].

While necessary, correlational studies of this kind are not sufficient [28–30]. Other types of evidence are required to support the existing hypotheses, to demonstrate and explain the presence of causality [2, 31], as well as to safeguard against type I errors in large-scale typological analyses (see detailed discussion in [32]). Possible complementary approaches include diachronic analyses of real data [33, 34], computational modeling [17, 35–38], and laboratory modeling [39–48].

The iterated learning model [49, 50] has been mentioned as a particularly promising approach for the investigation of the actual mechanisms of simplification and complexification [28, 51]. Recently, several experimental studies investigating various aspects of the mechanism using this method have been conducted [45, 47, 52–55]. However, the role of imperfect learning, which is often considered to be crucial [21], has seldom been focus of research. One important exception is the study conducted by Atkinson and colleagues [47] discussed in Sections 1.2 and 4.1.

Here we present a large-scale experiment which directly tests whether imperfect learning (in the absence of any other potential factors) simplifies linguistic production and whether this effect is amplified by iterated learning. Through this, we test whether imperfect learning shapes the distribution of morphological complexity across the languages of the world.

We discuss possible mechanisms of changes in complexity and the potential role of imperfect learning in Section 1.2 and formulate our research questions and predictions in Section 1.3. Section 2 describes the experiment. Section 3 presents the analysis of the results. We discuss the findings in Section 4 and conclude with Section 5.

## 1.2. Social structure and linguistic complexity

Of the five factors listed by Trudgill [11] (see Section 1.1), the two most studied ones are population size [18, 28, 40, 44, 47, 56] and level of contact. The potential connection between contact intensity or, specifically, between the relative proportions of adult and child learners in the population, and complexity decrease seems to be more straightforward and its descriptions are better fleshed out. Bentz and Winter [21] list three potential mechanisms of contact-induced case loss, which can be generalized to other instances of morphological simplification: (i) imperfect acquisition by adult learners; (ii) the tendency of native speakers to reduce morpho-syntactic complexity of their speech when talking to foreigners [47, 57]; (iii) the tendency of loan words to combine with more productive inflections, forcing the least productive ones out [58]. We focus on mechanism (i), which is the explanation most commonly entertained in the typological, sociolinguistic and evolutionary literature [28]. Unlike the other two mechanisms, it relies on the explicit assumption that imperfect learning directly causes simplification.

We understand imperfect learning as any kind of learning that results, for no matter what reason, in an inaccurate reproduction of the input language. It can be argued that natural language learning is always imperfect: language constantly changes in normal intergenerational transmission as well. In other words, language is never reproduced fully accurately. It makes sense then to consider to what degree the learning is imperfect and assume that if the degree is small (typical, for instance, in normal learning by children), it is not enough to cause simplification. If, however, the learning is imperfect to a large extent, which is typical, for instance, in non-native learning by adults [59], the simplification is likely to occur (see more about our operationalization of imperfect learning in Section 2.6). It is possible that there also exist qualitative differences between imperfect (e.g. non-native) and perfect or near-perfect (e.g. native) learning which influence whether simplification occurs or not, but this question is beyond the scope of the current article.

There is important corpus evidence in favour of the claim that adult learners simplify language (see, for instance [60]), but in-lab experimental tests are scarce. Atkinson et al. (Experiment 1) do show that imperfect learners simplify the morphology of an artificial language, especially in the early acquisition stages [47]. They do not, however, investigate the role of iterated learning (see Section 4.1 for further comparison of this study to ours). With this in mind, we perform a large-scale iterated-learning experiment that focuses on one primary question: can imperfect learning on its own, in the absence of any other factors, be a driving force in language change, in particular, morphological simplification?

## 1.3. The goals of the experiment

We ran an iterated learning experiment (see detailed description in Section 2). Artificial mini-languages were learned and transmitted further by members of 45 transmission chains, each consisting of 10 one-person generations. Each generation took the output language of the previous generation as the input, learned it, and reproduced it. This output was used as the input language for the next generation.

Some of the learners had less exposure to the language. We call them short-time learners (as opposed to long-time) and consider them as a model of imperfect learners (we perform a formal test of whether the exposure manipulation actually led to imperfect learning in Section 3.1). Importantly, we do not claim that in the real world imperfect learning is always caused solely by reduced learning time (see a more detailed discussion about the relation of our model to the real world in Sections 2.7 and 4.4).

15 chains represented a normal (N) condition (there are no short-time learners in the population). 15 chains represented a temporarily interrupted (T) condition (generations 2–4 are

short-time learners), 15 represented a permanently interrupted (P) condition (generations 2–10 are short-time learners). We use *interrupted* in the same sense as in Section 1.1: an interrupted language transmission is inhibited by some factors (in this case the presence of short-time learners).

The morphology of all initial input languages was the same and included number marking on nouns and additional agent-marking on verbs. We focus on one of the most prominent dimensions of complexity, viz. overspecification, that is, overt and obligatory marking of a semantic distinction that is not necessary for communication, following McWhorter's understanding [16]. Given our artificial setting we know exactly what information would be necessary for successful communication. It is limited to stems (names for entities and events) and number marking. The only instance of overspecification in our languages is the redundant agent-marking on verbs.

We make one prediction and ask two exploratory questions:

**Prediction**: *The complexity of languages will decline more in the interrupted conditions than in the normal one.* This decline will be a gradual accumulation of simplifications that occur in short-time-learner generations due to imperfect learning. Note that these individual simplifications may take place in some generations but not others, and when they take place, they may be small, but accumulating over time they will lead to a substantial change.

**Exploratory question 1**: *Which features are most likely to be affected by simplification?* And, if simplification occurs, can we identify factors which make features more or less vulnerable to it?

**Exploratory question 2**: *To what extent does the degree of overall simplification depend on how long the sequence of short-time-learner generations is?* To address this question, we model the "interruption" of transmission in two different ways: "temporary", with just three short-time-learner generations, and "permanent", with nine short-time-learner generations, and intend to compare the complexity trajectory in these two conditions.

We did not know how strong the effects we are interested in are. If the effects are small, we would need a large number of participants to make them visible. In order to achieve that, we went online.

## 2. Materials and methods

All data and code are available in S1 Appendix.

### 2.1. Participants and implementation

450 subjects (140 female, 310 male, mean age = 30.5, SD = 9.2) were recruited with the help of an advertisement placed in Russian online popular-science media (S1 Text). Subjects had to speak Russian natively, be at least 16 years old and not be a working as a linguist in order to participate. The experiment was conducted in accordance with the Norwegian Guidelines for research ethics in the social sciences, law and the humanities. The participants gave informed consent prior to the experiment, by ticking off a checkbox they had to confirm that they accept the rules and are willing to participate.

Before the start of the experiment, all participants were informed that at the end they would receive a number of randomly generated codes for a lottery. The participants were also informed that the number of codes they receive would depend on how successfully they learn the language, and so were encouraged to perform to the best of their abilities. After the training and test stages, *n+1* codes were generated for each participant, where *n* is the number of correct responses the participant gave in the comprehension test (see Section 2.5). When the experiment was over, four winning codes were randomly selected from all the generated

codes, and the participants who were issued these winning codes could receive an online book-store gift voucher worth 70 euros.

The webpage for the experiment was built using jsPsych JavaScript library [61]. See the supplementary material for further details (S1 Text) and a discussion of potential problems with running the experiment online (S2 Text).

## 2.2. Modeling the normal and the interrupted transmission

The experiment design was implemented using a version of the iterated learning model [49, 50], which is schematically illustrated in Fig 1. The iterated learning model approximates the diachronic development of language as a sequence of cultural transmissions between discrete generations of speakers: each generation took as the input the output language of the previous generation, learned it, and reproduced it. This output was used as an input language for the next generation.

We modeled iterated learning using transmission chains. Data from 45 language transmission chains, each 10 generations long and having one participant per generation, were collected. One third of the chains (1–15) were assigned to the normal condition (N), second third of the chains (16–30) to the temporarily interrupted condition (T), and the last third (31–45) to the permanently interrupted condition (P), see Fig 1. In the temporarily interrupted condition, generations two, three and four of the transmission chain had a reduced time to learn the language. In the permanently interrupted condition, all generations but the first had reduced learning time. We discuss our modeling approach in Section 2.7.

## Normal transmission

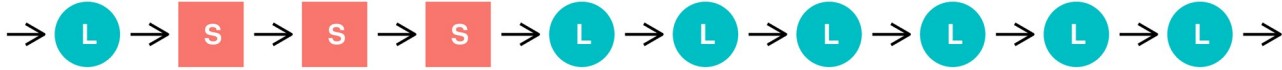

## Temporarily interrupted transmission

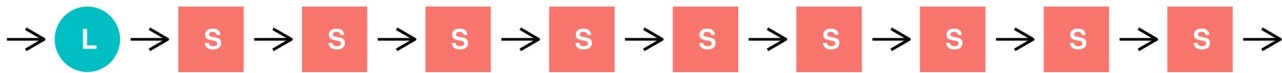

## Permanently interrupted transmission

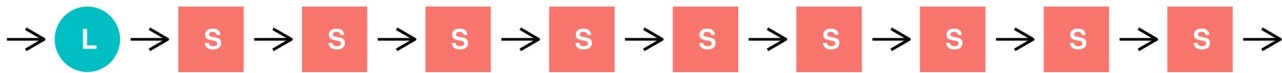

**Fig 1. The iterated learning model used in the experiment.** A schematic representation of the iterated learning model in the experiment and the difference between normal (a), temporarily interrupted (b) and permanently interrupted (c) chains. L denotes long-time learners, S denotes short-time learners.

## 2.3. Initial input language structure

A total of 15 languages were created as input languages for generation 1 participants (we label them as generation 0 languages), each of them being used once in every condition. Each language had the morphological structure outlined in Fig 2, and contained two noun stems for different agents, a noun ending for marking plurality, three verb stems for different events and a verbal ending that also marked the agent. This double agent-marking can be viewed as an extremely simple model of gender agreement. The system resembles the more complex Russian morphosyntactic system where nouns are marked for number, and adjectives and verbs agree with nouns in number and gender. What is important is that agreement is salient and pervasive in Russian morphosyntax, and thus the participants' mother tongue was not imposing the pressure to shed the redundant agent-marking on them.

All sentences in each language mapped transparently onto its meaning space, as shown in Fig 2. All input languages had the same structure: CVC for nominal stem, C for nominal ending, C for verbal stem, V for verbal ending. It is common in iterated learning studies to seed the transmission chains with different, but structurally isomorphic languages in order to exclude influence of any potential idiosyncrasies of a particular language. See the supplementary material (S3 Text) for a more detailed description of creating the initial input languages.

Throughout the article, we will consider the first word in the sentence to be a noun, the second (if it is present) to be a verb. Manual inspection shows this is a reasonable analysis.

It is possible that participants sometimes analyzed the languages not in the same way as we do here. Feedback from pilot participants, for instance, indicated that what we call verbs they sometimes perceived as adjectives. It was also possible that they segmented the words differently (or did not segment them at all), or invented different rules to explain the vowel change in "verbs". These and other potential discrepancies do not affect the results and their interpretation in any way, since we are interested in the structure of language change and not in the learners' perception of it. We offer the analysis in Fig 2 just as a convenient tool of describing the languages.

## 2.4. Training stage

At the very beginning of the experiment participants were randomly assigned to a new generation in one of the transmission chains, after which they were presented with the instructions. The instructions framed the experiment as a part of an expedition trying to establish contact with and learn the language of an alien race of planet Epsilon with the help of an Epsilonian named Seusse. The participants were encouraged to learn the language and produce the answers to the tasks even if they were not completely sure about their correctness.

After the initial instructions (S1 Text), the participants encountered a series of learning blocks, during which they were presented with all sixteen possible sentences of the alien language in a random order accompanied by the corresponding pictures. Each sentence was presented on the screen for four seconds. Learning blocks were interspersed with interim tests in which the participants were consequently presented with eight randomly selected pictures and were asked to type in the corresponding words in the alien language for the pictures. The participants were forbidden to take any notes and did not receive any feedback during interim tests. The number of learning blocks, and consequently the number of interim tests, differed between long-time-learner generations and short-time-learner generations: long-time learners received six blocks of training, whereas short-time learners received three. Short-time learners were generations 2–4 in condition N (chains 16–30) and generations 2–10 in condition P (chains 31–45).

|  |  | agent:<br>round animal | agent:<br>square animal |
|---|---|---|---|
| event:<br>none | number:<br>singular | $seg_N$ | $fuv_N$ |
|  | number:<br>plural | $seg_N\text{-}l_{PL}$ | $fuv_N\text{-}l_{PL}$ |
| event:<br>fall apart | number:<br>singular | seg m-o | fuv m-i |
|  | number:<br>plural | seg-l m-o | fuv-l m-i |
| event:<br>grow antlers | number:<br>singular | seg r-o | fuv r-i |
|  | number:<br>plural | seg-l r-o | fuv-l r-i |
| event:<br>fly | number:<br>singular | seg b-o | fuv b-i |
|  | number:<br>plural | seg-l b-o | fuv-l b-i |

**Fig 2. Meaning space of the artificial languages.** The meaning space of generation 0 language from chain 1 with the corresponding sentences. Morphemes are hyphenated for clarity's sake (hyphens were absent in the actual languages that the participants saw).

## 2.5. Testing stage: Comprehension and production tests

In the last part of the experiment, the participants were presented with all sixteen pictures one by one along with the text "Describe this picture in Epsilon so that Seusse could understand you" and were asked to type in the sentence that in their opinion corresponded to the picture. The order of the pictures was randomized. The resulting output language was used as the input language for the following generation of the chain.

The participants were also given a comprehension test in order to determine the number of their prize codes. We do not report on the test for brevity's sake (see S4 Text for its description, S5 Text and S1 Fig for the results). The participants did not receive any feedback in either of the tests.

## 2.6. Transmission fidelity and imperfect learning

In order to estimate whether the learning is imperfect or not (cf. discussion in Section 1.2), we measure transmission fidelity, i.e. transmission error subtracted from 1. Transmission error is the average of pairwise normalized Levenshtein distances between signals that correspond to the same meaning (i.e. the same picture) in the input and the output language of a participant (cf. [50, 62]), see S3 Fig for detailed results.

Even though pilot experiments indicated that six blocks are enough for the participants to learn the language perfectly, many participants did not manage to learn the language fully with six blocks in the study reported here. Thus, long-time learners were not "perfect" learners, and the conditions differed in the degree of imperfect learning rather than its presence or absence. In Section 3.1 we show, however, that there was a large difference in the degree of imperfect learning between the normal condition and the interrupted ones, which means that model satisfies our needs.

## 2.7. Modeling approach and model validity

Our model was simple. Perhaps most saliently, imperfect learning was achieved solely by manipulating learning time. The simplicity of the design is intentional: it allows to evaluate whether imperfect learning on its own, in the most basic case, could lead to language simplification and allows our model to serve as a baseline for the evaluation of more elaborate models in future studies. We do not claim that insufficient learning time is the only reason for imperfect learning in the real world.

Further, each generation consisted of one participant only and there was no real communication task included, while some studies suggest that communication is important factor that exerts pressures on the structure of language [62, 63]. However, the experiment is not supposed to be an ecologically valid model of the complex processes going on in the real world (language contact, non-native acquisition, etc.). It is, nevertheless, supposed to be an internally valid model of influence exerted on language change by imperfect learning. We investigate language change in the lab to evaluate whether the hypothesized mechanism is possible in the real world.

While we acknowledge that the languages and the process of learning (and the reasons for its imperfection) may be different in the lab and the outside world in some aspects, following a large literature in evolutionary linguistics, we make the plausible assumption that the fundamental mechanisms of change are nevertheless shared in these two situations. We refer a skeptical reader to the supplementary material (S6 Text), where we discuss other simplifications that our model is built on.

## 2.8. Measuring complexity

Even though we use a narrow operationalization of complexity (overspecification, see Section 1.3), it can still be quantified in different ways. We focus on the fact that overspecification leads to higher number of distinct word forms, a property which Bentz et al. [22] refer to as lexical diversity. We will call it lexicogrammatical diversity, since it depends not only on the number of different lexemes in a text, but also on a number of different word forms. This property can be captured by calculating type-token ratio, or TTR. TTR is defined as the number of distinct words (types) in the language divided by the total number of words (tokens). A word is understood as a sequence of letters delimited by white spaces or other non-word characters. We did not perform any lemmatization, i.e. *mi*, *mo*, *seg* and *segl* in language 1–0 (see Fig 2) are four different words. See S7 Text for more details.

Given that there are different means of measuring complexity, each with its own advantages and drawbacks [64], we would like to motivate our choice of measure. TTR is a simple, easily interpretable and reproducible measure, which does not require elaborate theoretical assumptions. It is usually applied to corpora, but given the nature of our artificial languages, it is an adequate measure of their lexicogrammatical diversity. First, by design, each language is a complete enumeration of all possible meanings and can be construed as a corpus. Second, the distribution of meanings in the Epsilon universe is always uniform, i.e. we do not have to worry about the potential effect of frequency distributions influencing the measure. Finally, TTR is highly sensitive to text size, but since all our languages share the same meaning space, they can be treated as parallel corpora, which resolves the problem. Simplification should then result in the loss of overspecification, i.e. lower TTR.

Bentz et al. [22] describe other measures of lexicogrammatical diversity (Shannon entropy and Zipf-Mandelbrot's law parameters), but mention that TTR is the most responsive of these three, which is important given the small size of our "corpora". We make an additional measurement using entropy, which yields similar results (see S8 Text and S4 Fig).

## 3. Results

In Section 3.1, we show that the assumption our experiment is based upon is valid and that reduced learning time did lead to imperfect learning. In Section 3.2, we show that imperfect learning led to a decrease in overspecification. In Section 3.3, we investigate this decrease more closely and show that it affected verbs, but not nouns, and that within verbs the endings (agent markers) were affected much more strongly than the stems (lexical meanings).

## 3.1. Reduced learning time leads to imperfect learning

We start by testing the assumption that reduced learning time actually leads to imperfect learning (see Section 2.2). The differences between transmission fidelity at generation 2 in the normal condition (only long-time learners) and both interrupted conditions (only short-time learners) are represented on Fig 3, and a two-tailed *t*-test yields the following results: $t(42.9) = 2.84$, $p = 0.007$, 95% CI for difference in means [0.01, 0.06], Cohen's $d = 0.73$. We do not include later generations into analysis since their learner type is confounded with the complexity of the input, which depends on the output of previous generations. See S3 Fig for more detailed results.

## 3.2. Imperfect learning leads to simplification

The change of overall TTR over time is represented in Fig 4.

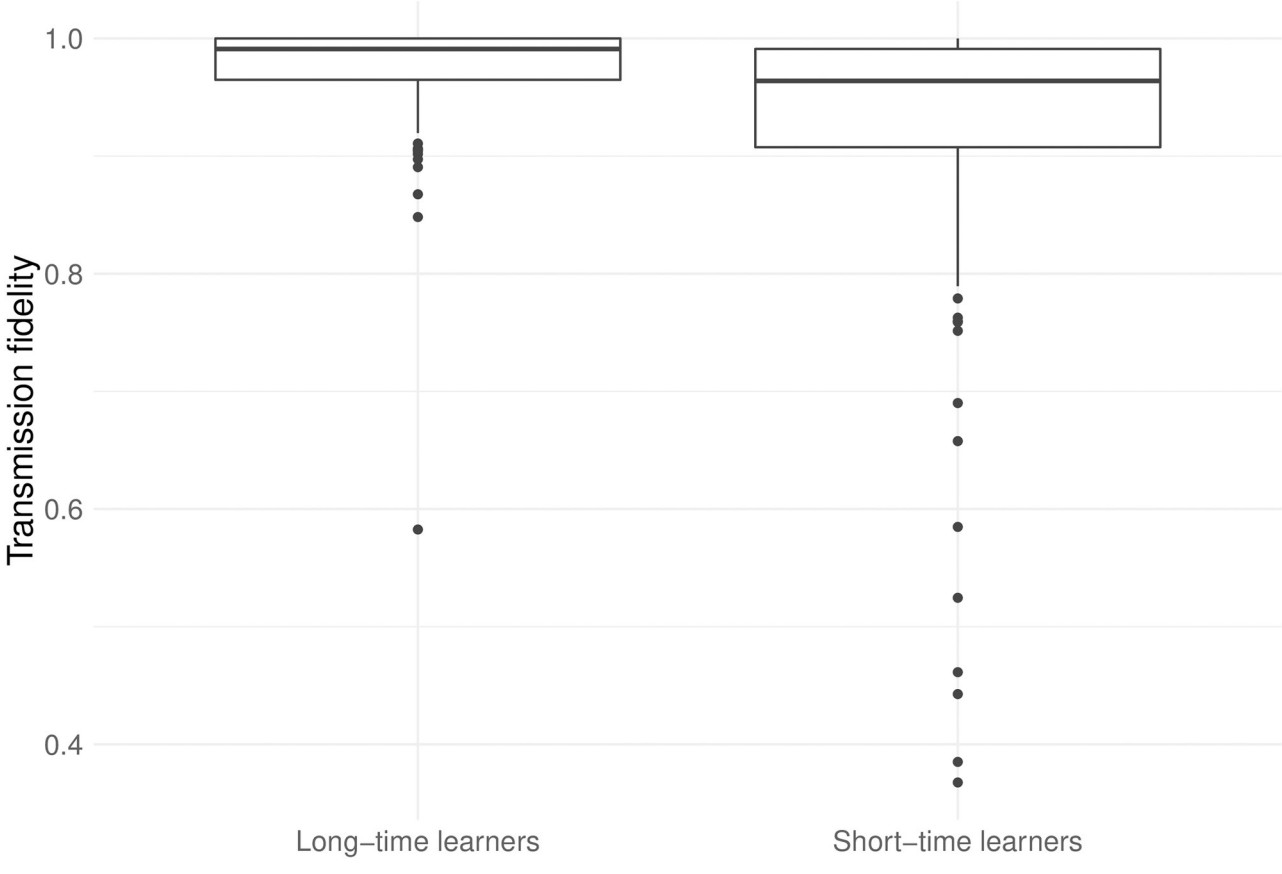

**Fig 3. Transmission fidelity at generation 2.**

In order to explore the role of condition and generation we fit a linear mixed-effect regression model (LMM). We largely follow the recommendations for applying regression models outlined in [65]. We do all calculations in R [66], using packages `lme4` [67] for constructing mixed-effect models and `lmerTest` [68] for calculating significance of estimated parameters by REML *t*-tests with the Satterthwaite approximation to degrees of freedom. We also use `ggplot2` [69] for creating plots and `effsize` [70] for measuring effect sizes for *t*-tests. R scripts with comments are available in S1 Appendix.

The LMM includes fixed effects of generation, condition and their interaction, and by-chain random intercepts and random slopes for generation (the `lme4` notation is provided in Eq 1). We use treatment coding (a.k.a. dummy coding) for condition, with condition T as reference level. Since TTR is on a bounded scale (0, 1], we log-transform the TTR values before fitting the model. See S1 Appendix for the R implementation and tests of the assumptions.

$$ttr \sim condition * generation + (1 + generation|chain) \tag{1}$$

The summary of the model is given in Table 1.

From Table 1 we can conclude that there was a reduction of TTR over generations in condition T (since the negative slope for generation is significantly different from zero), and a similar reduction in condition P (since the interaction for generation and condition P is small and not significant). In condition N, however, the reduction was smaller, since the interaction between condition N and generation is of the same magnitude as the effect of generation.

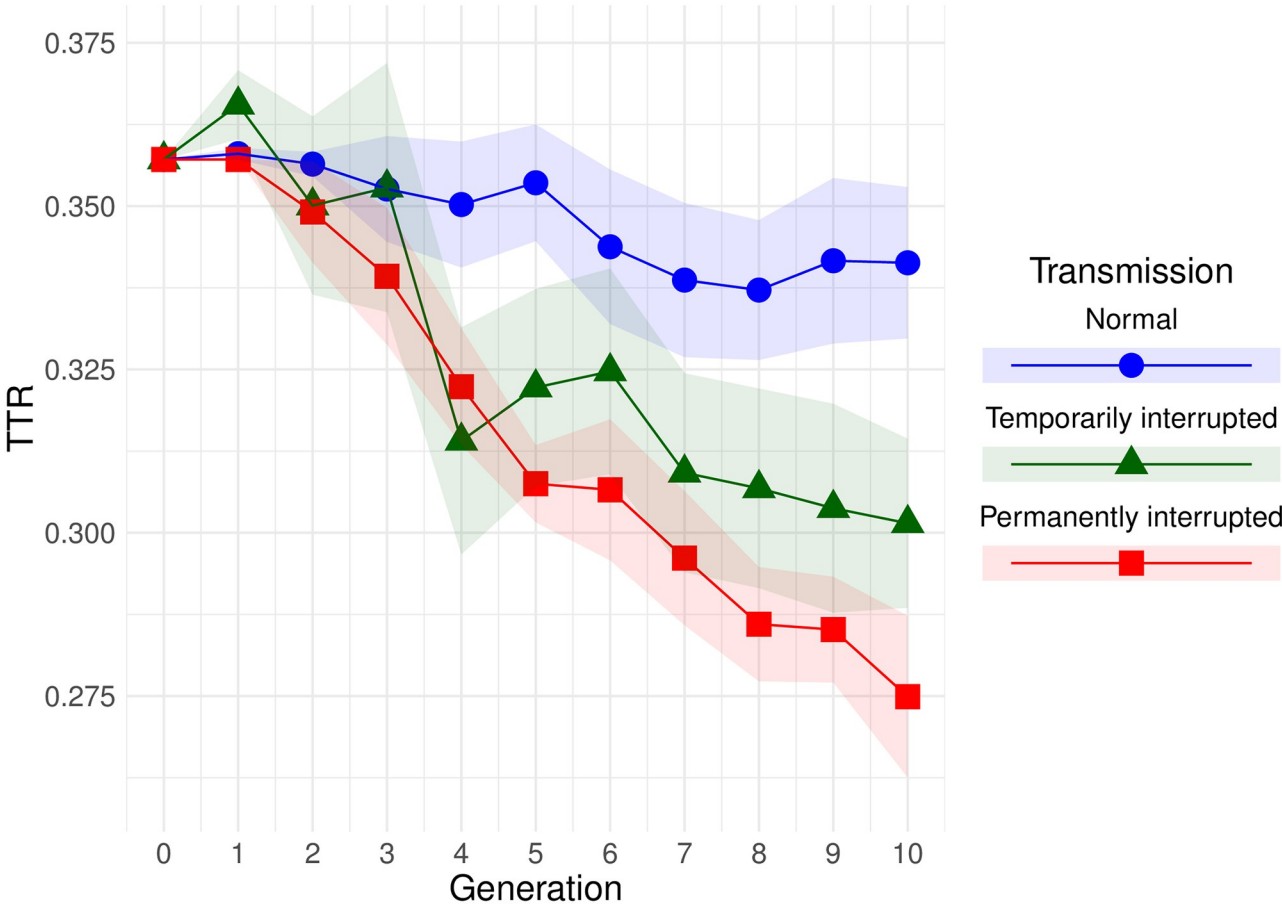

**Fig 4. Change of type-token ratio over time.** Shaded bands show standard error.

Interestingly, if we compare TTR of long-time and short-time learners at generation 2 (see Fig 5), as we did in Section 3.1 with transmission fidelity, we observe no differences in means, though variance is visibly different between the conditions ($t(32.3) = 0.86$, $p = 0.395$, 95% CI for difference in means [-0.009, 0.023], Cohen's $d = 0.196$). In other words, imperfect learning does not necessarily cause simplification immediately, within one generation.

### 3.3. Simplification primarily affects agent-marking on verbs

Fig 6 represents TTR calculated separately for nouns and verbs. For verbs the pattern of change is similar to the overall trend, cf. Fig 4. For nouns, no decrease is observed (there is a very small increase, but it is not significant).

**Table 1. Model summary: Type-token ratio as predicted by generation and condition.**

| Fixed effect | Estimate | SE | $t$(df = 42) | $p$ |
|---|---|---|---|---|
| Intercept | -1.024 | 0.017 | -59.91 | $<1 \times 10^{-15}$ |
| Generation | -0.021 | 0.005 | -4.70 | $<2.8 \times 10^{-5}$ |
| Condition N | -0.0004 | 0.024 | -0.01 | 0.989 |
| Condition P | 0.017 | 0.024 | 0.44 | 0.661 |
| Generation x Condition N | 0.014 | 0.006 | 2.19 | 0.034 |
| Generation x Condition P | -0.008 | 0.006 | -1.28 | 0.207 |

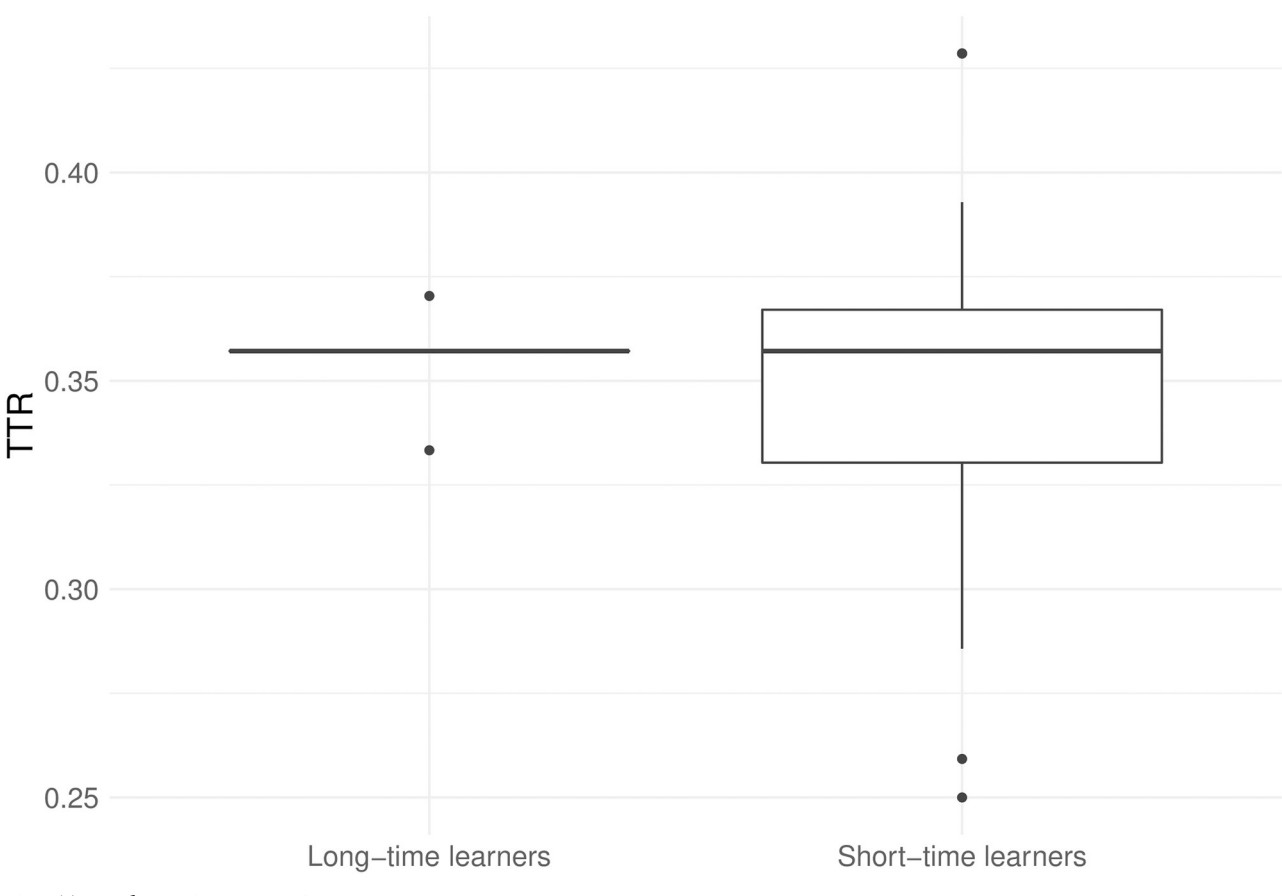

**Fig 5. Type-token ratio at generation 2.**

We fit an LMM with a specification similar to Eq 1, but add part-of-speech (reference level: noun) as a fixed effect. The model includes all possible interactions (that is, three two-way interactions and one three-way interaction). With the maximal random-effect structure, the model does not converge. We deal with that by removing the correlation parameter (cf. [71]). The resulting specification in lme4 notation is shown in Eq 2.

$$ttr \sim condition * generation * pos + (0 + generation|chain) + (1|chain) \qquad (2)$$

The summary of the model is given in Table 2.

The most interesting coefficients in Table 2 are those that include the effect of the generation. For nouns in condition T, there is a minor increase in complexity (though the *p*-value is higher than any conventional significance threshold, and thus we do not have strong evidence to claim that the true value of the coefficient is different from zero), the same holds for other conditions. For verbs, however, the effect of generation is reversed and clearly negative (as was the case for the TTR in general). The slope is less steep in condition N.

To sum up, verbs got simpler, while nouns did not. There was a clear difference between the normal condition and the interrupted ones.

We resort to manual analysis in order to qualitatively explore how exactly languages may be simplified and complexified. Here and below we will refer to languages by means of one letter (N for normal chains, T for temporarily interrupted, P for permanently interrupted) and two

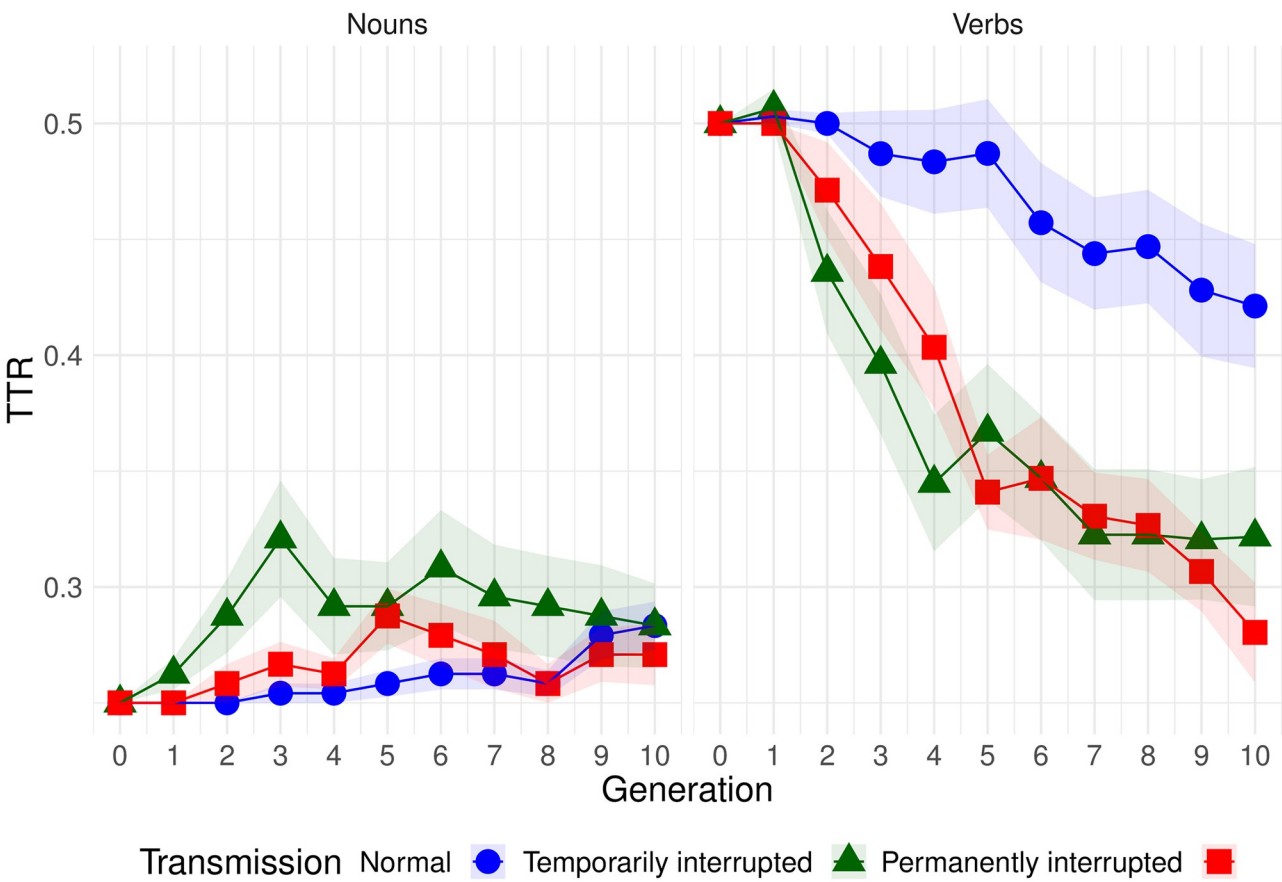

**Fig 6. Change of type-token ratio over time separately for nouns and verbs.** Shaded bands show standard error.

**Table 2. Model summary: Type-token ratio as predicted by generation, condition and part of speech.**

| Fixed effect | Estimate | SE | t(df) | p |
|---|---|---|---|---|
| Intercept | -1.301 | 0.027 | -48.74 (219) | $<1 \times 10^{-15}$ |
| <u>Generation</u> | 0.006 | 0.006 | 1.08 (128) | 0.280 |
| Condition N | -0.102 | 0.038 | -2.70 (219) | 0.007 |
| Condition P | -0.063 | 0.038 | -1.66 (219) | 0.098 |
| POS verb | 0.549 | 0.037 | 14.84 (917) | $<1 \times 10^{-15}$ |
| <u>Generation x Condition N</u> | 0.004 | 0.008 | 0.53 (128) | 0.600 |
| <u>Generation x Condition P</u> | 0.000 | 0.008 | -0.05 (128) | 0.963 |
| Generation x POS verb | -0.060 | 0.006 | -9.53 (917) | $<1 \times 10^{-15}$ |
| Condition N x POS verb | 0.190 | 0.052 | 3.64 (917) | $2.9 \times 10^{-4}$ |
| Condition P x POS verb | 0.137 | 0.052 | 2.62 (917) | 0.009 |
| <u>Generation x Condition N x POS verb</u> | 0.026 | 0.009 | 2.98 (917) | 0.003 |
| Generation x Condition P x POS verb | -0.009 | 0.009 | -1.04 (917) | 0.297 |

*Note*: The most important effects are underlined.

**Table 3. Structure of first and final generation artificial languages in chain N9.**

| Glosses | | Gen 0 | | Gen 10 | |
|---|---|---|---|---|---|
| | | round animal | square animal | round animal | square animal |
| - | sg | rub | vad | rub | vad |
| | pl | rubp | vadp | **rubs** | vadp |
| fall apart | sg | rub le | vad lo | rub le | vad lo |
| | pl | rubp le | vadp lo | **rubs** le | vadp lo |
| grow antlers | sg | rub ze | vad zo | rub ze | vad zo |
| | pl | rubp ze | vadp zo | rubp ze | vadp zo |
| fly away | sg | rub ne | vad no | rub ne | vad no |
| | pl | rubp ne | vadp no | rubp ne | vadp no |

*Note*: Differences between nouns are marked in bold.

numbers (a-b), where a is the number of the chain (ranging from 1 to 45), and b is the number of the generation (ranging from 0 to 10).

Two examples of the complexification of the nominal system can be found in languages N9–10 (Table 3) and T18–10 (Table 4).

N9–10 has two patterns of marking nominal number: -*p* (the main one) and -*s*. The -*s* ending originally emerged as a random mutation at generation 3 in a single sentence ('round animals fall apart') and was preserved unchanged (which is possible due to high transmission fidelity) until generation 10, where it spread also to the sentence 'round animals', thus developing from a single exception into a minor pattern.

Language T18–10 lost all double agent-marking, and had its nominal system reorganized, with an emergent pattern where number distinction is marked through non-concatenative morphological processes—metathesis for one noun (*senz*, *sezn*) and consonant mutations for another (*sign*, *dign*). These changes, however, are not instances of complexification according to our definition and will not be captured as such by the TTR measure. The mutated plural form *digm* (instead of *dign*, a random change first appearing at generation 8), however, would.

This language deserves further attention. Its unique development emerged through several stages (see chain T18 in S1 Appendix). First, a poor learner in generation 3 drastically reorganized the system, introducing numerous inconsistencies. Through generations 4–7, these inconsistencies were either eliminated or underwent exaptation (cf. [72]), which resulted in a stable system at generation 8 (identical to that in generation 10).

**Table 4. Structure of first and final generation artificial languages in chain T18.**

| Glosses | | Gen 0 | | Gen 10 | |
|---|---|---|---|---|---|
| | | round animal | square animal | round animal | square animal |
| - | sg | dig | sez | **senz** | **sign** |
| | pl | dign | sezn | **sezn** | **dign** |
| fall apart | sg | dig mo | sez mu | **senz** *po* | **sign** *po* |
| | pl | dign mo | sezn mu | **sezn** *po* | **dign** *po* |
| grow antlers | sg | dig po | sez pu | **senz** *ho* | **sign** *ho* |
| | pl | dign po | sezn pu | **sezn** *ho* | **digm** *ho* |
| fly away | sg | dig ho | sez hu | **senz** *mo* | **sign** *mo* |
| | pl | dign ho | sezn hu | **sezn** *mo* | **dign** *mo* |

*Note*: Differences between verbs in the initial and the final language are marked in italic, differences between nouns are marked in bold.

**Table 5. Structure of first and final generation artificial languages in chain T25.**

| Glosses | | Gen 0 | | Gen 10 | |
|---|---|---|---|---|---|
| | | round animal | square animal | round animal | square animal |
| - | sg | jal | rok | jal | rok |
| | pl | jald | rokd | jald | rokd |
| fall apart | sg | jal bu | rok be | jal *te* | rok *te* |
| | pl | jald bu | rokd be | jald *te* | rokd *te* |
| grow antlers | sg | jal fu | rok fe | jal fu | rok *fu* |
| | pl | jald fu | rokd fe | jald fu | rokd *fu* |
| fly away | sg | jal tu | rok te | jal *fe* | rok *fe* |
| | pl | jald tu | rokd te | jald *fe* | rokd *fe* |

*Note*: Differences between verbs in the initial and the final language are marked in italic.

For verbs, the manual analysis shows that the decrease in diversity occurred primarily due to the loss of the double agent-marking, either partial or full. T25–10 (Table 5) is an example of a language where the double agent-marking has completely disappeared. Interestingly, this language did not just abandon one of the agent markers -*e* and -*u* in favour of another one, but instead kept both, reanalyzing them as parts of the stems (out of 14 languages that shed the double agent-marking completely, only three abandon one of the markers, another 11 reanalyze them). Thus, verbs *fu* and *fe* both originate from the generation zero stem *f-*, while the stem *b-* did not survive.

To test the aforementioned claim that the complexity loss mostly affects agent-marking (expressed by the last letter of the verb, when present), but not the lexical meaning (usually expressed only by the first letter), we calculate the TTR of verb "stems" (first letters) and verb "endings" (last letters). To make the measurement more adequate, we perform an additional manipulation.

For endings, we calculate TTR within every verb and then average them. The reason for this step in calculations is that we want to focus on agent-marking and thus eliminate other semantic factors that could inflate TTR. If there is no agent-marking, the same verb should always look the same, and the TTR should be 0.25. For example, for language T25–0 (Table 2) that means averaging the TTR over the three subcorpora that all look like {u, u, e, e}, resulting in the value of 0.5. For language T25–10, the subcorpora look like {u, u, u, u}, {e, e, e, e}, {e, e, e, e}, and the resulting average TTR is 0.25. We should note that in some languages the ending gets reanalyzed and denotes not the type of agent, but the number of agents. We consider this phenomenon to be a type of agreement with subject, equally complex to the double agent-marking present in the initial languages, and thus our TTR measure reflects it correctly.

For stems, we calculate TTR within two subcorpora: verbs that occur with the noun denoting the round animal and verbs that occur with the noun denoting the square animal. The rationale is the same as for endings: we want to eliminate all differences between verbs apart from lexical meaning. The drawback of this method is that languages like T25–10, where two verbs have the same first letter (but still have different stems since the vowel has been reanalyzed as part of the stem) receive a lower TTR than they should. Both subcorpora look like {t, t, f, f, f, f} and the TTR is 0.33, while 0.5 would have been a more adequate value. Such cases, however, are rare.

For further details of TTR calculation, see S7 Text.

The change of TTR of stems and endings over time is represented on Fig 7. We fit an LMM with the same specification as in Eq 2, but instead of part of speech, we add morpheme type

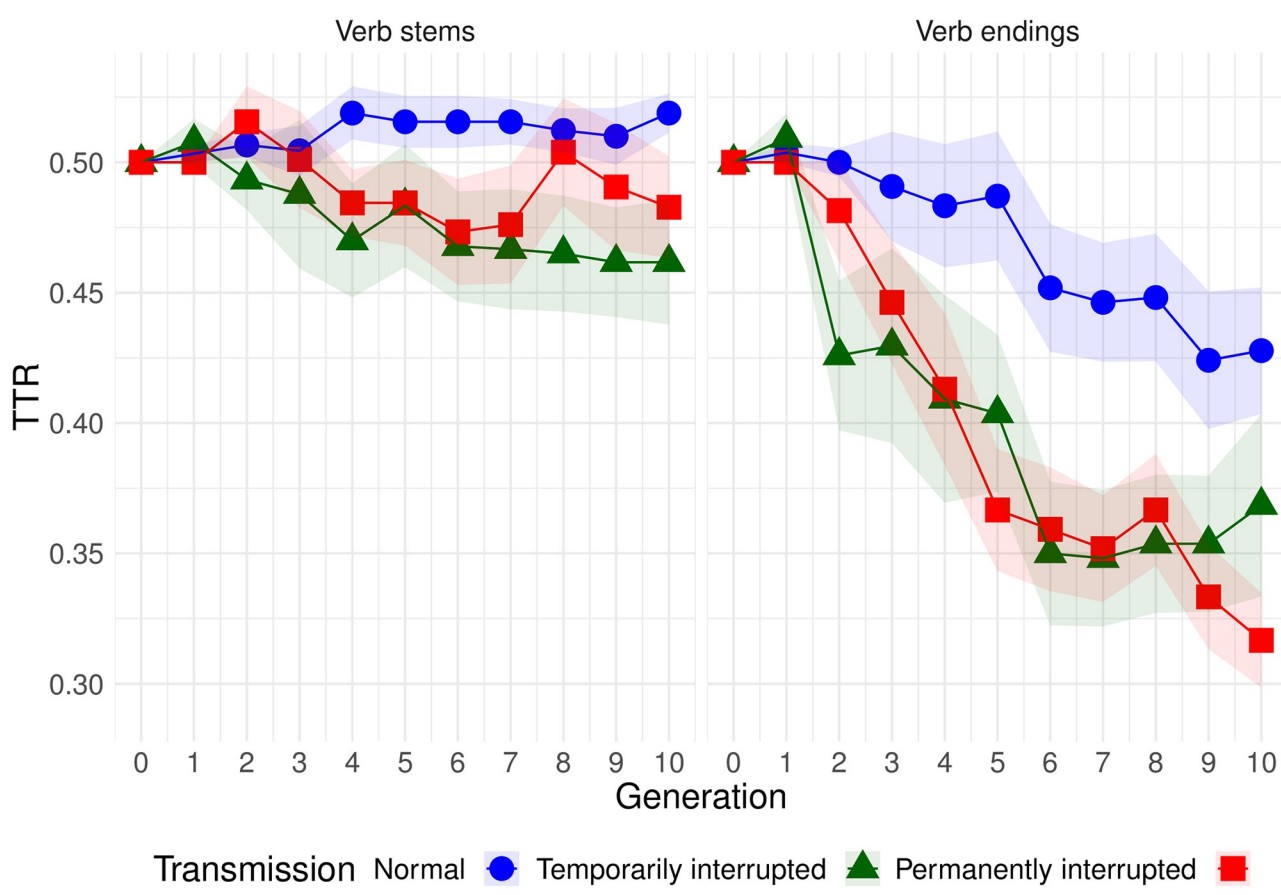

**Fig 7. Change of type-token ratio over time separately for verb stems and verb endings.** Shaded bands show standard error.

(stem or affix, with stem being the reference level) as a fixed effect. The model is applied to verb data only. The summary of the model is given in Table 6.

The most important pattern that can be observed is that complexity decreased over time in condition T, and that this trend was much more pronounced for affixes than for stems. In condition N, the trend was weaker (absent for stems).

## 4. Discussion

### 4.1. Languages simplify more in conditions with interrupted transmission and imperfect learning

Our **Prediction** (*the complexity of languages will decline more in the interrupted conditions than in the normal one*) is supported by the results: the complexity (narrowed down to over-specification and measured as type-token ratio) clearly decreased over generations in both interrupted conditions, whereas in the normal condition, the slope was less steep and the decrease was very small (see Section 3.2).

As expected, the simplification was gradual (at generation 2, for instance, there was no significant difference between the normal and the interrupted conditions). In other words, it was not normally the case that a single generation simplified the language dramatically. The difference between the output languages of generations *n* and *n+1* was usually small, and some of the small changes that individual speakers make eventually led to simplification.

**Table 6. Model summary: Type-token ratio as predicted by generation, condition and part of speech.**

| Fixed effect | Estimate | SE | t(df) | p |
|---|---|---|---|---|
| Intercept | -0.695 | 0.026 | -26.57 (122) | $<1 \times 10^{-15}$ |
| Generation | -0.011 | 0.006 | -2.01 (84) | 0.048 |
| Condition N | 0.008 | 0.037 | 0.23 (122) | 0.820 |
| Condition P | 0.004 | 0.037 | 0.11 (122) | 0.910 |
| Morpheme Affix | -0.050 | 0.032 | -1.58 (911) | 0.114 |
| Generation x Condition N | 0.014 | 0.008 | 1.75 (84) | 0.085 |
| Generation x Condition P | 0.006 | 0.008 | 0.70 (84) | 0.487 |
| Generation x Morph Affix | -0.030 | 0.005 | -5.56 (911) | $3.5 \times 10^{-8}$ |
| Condition N x Morph Affix | 0.070 | 0.045 | 1.57 (911) | 0.116 |
| Condition P x Morph Affix | 0.052 | 0.045 | 1.15 (911) | 0.249 |
| Generation x Condition N x Morph Affix | 0.005 | 0.008 | 0.70 (911) | 0.487 |
| Generation x Condition P x Morph Affix | -0.015 | 0.008 | -1.98 (911) | 0.048 |

*Note*: The most important effects are underlined.

Since the only difference between the conditions was the presence of short-time learners, we can claim that simplification was caused by reduced learning time. Significant differences in transmission fidelity at generation 2 indicate that reduced learning time causes imperfect learning (see Section 3.1). We would like to reiterate that we do not claim that in the real world imperfect language learning occurs solely due to reduced learning time. In our study, reducing learning time is just a technical means of ensuring imperfect learning, which we consider the real cause of morphological simplification.

The fact that a small decrease was observed in condition N, too, is unsurprising, given that the learning in this condition was not entirely perfect either (see S3 Fig). As we mentioned in Section 2.4, the difference between conditions was in degree of imperfect learning, not its presence/absence.

It is interesting to compare our results to Experiments 1 and 2 by Atkinson et al. [47]. In Experiment 1, adult learners were trained on a morphologically complex miniature language and had to reproduce it. At the early stages of learning, the output languages had noticeably simpler morphology. Complexity, however, increased as the participants had more time to learn the language and approached that of the original languages at the later learning stages. There was no intergenerational transmission.

Experiment 2 investigated the propagation of simplifications through subsequent learning and had a more complex design. The languages produced by the participants of Experiment 1 were used as the input for a second generation of learners. Two parameters of these input languages were manipulated: whether they came from two participants of Experiment 1 or eight participants and whether they consisted of "complex" languages only or of the mix of "complex" and "simple" languages. Complex languages were those produced at the final learning stage (approaching the original language), simple were those produced at an early learning stage. Atkinson et al. conclude that neither the population size nor the complexity of the input languages affected the complexity of the output languages and offer several possible explanations for this finding. Note that the purpose of Atkinson et al.'s Experiment 2 (and 3, not discussed here) was to solve the problem of linkage: how can individual-level simplifications spread to the whole population. We do not address the problem of linkage in this study.

Taken together, Atkinson et al.'s Experiment 1 and our study suggest that imperfect learning on its own (especially if it is amplified by iterated learning) does cause simplification.

Atkinson et al.'s Experiment 2, however, suggests that this effect may be moderated by other factors.

## 4.2. Redundant agent marking is most affected by simplification

With respect to our **Exploratory question 1** (*which features are most likely to be affected by simplification?*), the results clearly demonstrate that the decrease in complexity for the double agent-marking (verbal ending) was stronger than for any other feature. See S9 Text and S5 Fig where an additional finer-grained analysis shows the same trend even more clearly.

We hypothesize that there are three possible explanations for that. First, this feature is redundant and learning it is not necessary to preserve the expressive power of the language. Redundant features are more likely to be eliminated, see [30, 73]. Since the experiment does not involve a communication task, there is actually no particular pressure for expressivity (apart from the general incentive to reproduce the input language as accurately as possible). However, the comprehension test (see S4 and S5 Texts) was framed as a dialogue with the friendly Epsilonian Seusse, and the purpose of it was to create the impression of communication.

Second, this feature is more complex than others, since it involves a long-distance dependency (between the stem of the first word and the affix of the second one) and learning it may potentially be more difficult [59].

Third, there is a range of other properties that could all be categorized under the label of "salience" (not to be confused with salience in the sociolinguistic sense). In the input languages, the verbal ending always comes last in the sentence, it is short and consists of a single vowel (though note that consonantal verbal stems and nominal endings are not longer), and it occurs in 12 sentences out of 16. These properties are mostly preserved across generations.

It can be argued that it makes as much sense to label nominal stem as redundant agent-marking as verbal ending, since they both denote agent. The reasons why it is the verbal ending which gets eliminated are probably greater learning difficulty and lesser salience.

Unlike verbs, nouns do not get simplified. If anything, according to the TTR measure, they become slightly more complex (see Section 3.3), but we cannot claim that this effect is robust and reproducible. This observation, however, may deserve to be further tested in future studies in the light of different hypotheses considering complexification in real languages [14, 18, 74, 75].

## 4.3. No evidence that simplification is strongly affected by the degree of interruption in transmission

As to our **Exploratory question 2** (*to what extent does the degree of overall simplification depend on how long the sequence of short-time-learner generations is?*), there were no strong differences between the two interrupted conditions, i.e. we find no evidence that the number of short-time-learner generations mattered in our setting.

One possibility is that once the process of simplification is started by the three short-term-learner generations, it will be continued by subsequent generations regardless of their learning time. In other words, long-time learners reproduce the initial languages with complex, but consistent structure rather faithfully, but continue to simplify input languages with inconsistent structure (which may be less overspecified, but also more irregular) [76].

Another possibility is that there actually is a difference between conditions T and P and, given longer chains or more chains per condition, we would have seen temporarily interrupted chains reach a plateau after the interruption, while permanently interrupted chains would

have continued the downward trend. Fig 4 suggests it might be true, but from our dataset we cannot conclude whether the effect is real.

### 4.4. Relation to the real world

While the experiment was not designed to model a specific case of natural language simplification, it is nevertheless useful to note that broadly similar patterns of reduction or loss of overspecified features have been claimed to follow language contact involving high numbers of L2 learners [77]. For example, many cases of overspecified features reconstructed for Proto-Germanic that are retained in most modern Germanic languages, e.g. gender marking on the article, (redundant) inherent reflexive marking, or use of both 'have' and 'be' as auxiliaries for marking perfect aspect, have been lost in English. It has been argued that these changes were initiated by close contact between Old English and Old Norse speakers following Scandinavian invasion of England in the late ninth and early tenth centuries [78]. Similarly, [16] argues that Mandarin Chinese, Persian, some colloquial Arabic varieties, and Malay all display lower levels of morphological complexity and overspecification compared to related languages and links these developments to previous episodes of close language contact.

More specifically, in Section 2.3, we claimed that double agent-marking in our languages can be viewed as an extremely simple model of gender agreement. Agreement is often considered to be redundant in natural languages too [11, 74]. While it is clear that repeating information can be beneficial in noisy channels, and while there is evidence that agreement has certain functions in language processing [79, 80], our point is that it is not necessary for languages to have a special device to perform these functions, they can operate equally well without it. Importantly, imperfect learning caused by language contact has been claimed to be a key factor in disappearance of agreement [77].

A word of caution is in order. While our experiment can reveal cognitive biases, they are not the only factor in language change. Social factors interact with cognitive ones in complicated ways, amplifying or masking them [53–55, 81]. Our data are supportive of the hypothesis that imperfect learning contributes to the elimination of overspecification. It seems, however, that imperfect learning alone is unlikely to eliminate overspecification completely. We cannot exclude the possibility that our ten generation chains were too short, and over a longer period of time we would have seen more cases of complete simplification, but it is also possible that other factors (e.g. presence of regularizing learners, e.g. children, and/or favorable social conditions) have to be present.

### 4.5. Visual summary

Finally, we would like to summarize our claims in a causal graph in the format of the CHIELD database [82], see Fig 8.

Representing causal hypotheses about language change by causal graphs is becoming increasingly popular in evolutionary linguistics and related fields and has several important benefits. First, it is a convenient visual way to explicitly summarize the causal claims, showing what kind of evidence (if any) supports every claim. Second, causal graphs are machine-readable and thus can be easily accumulated in a single database (see S1 Appendix for a machine-readable representation), which can become a tool for expressing, exploring and evaluating hypotheses.

## 5 Conclusion

The fields of cognitive science, linguistic typology, sociolinguistics, language evolution, among others, are engaged in an ongoing discussion on whether social factors, such as language

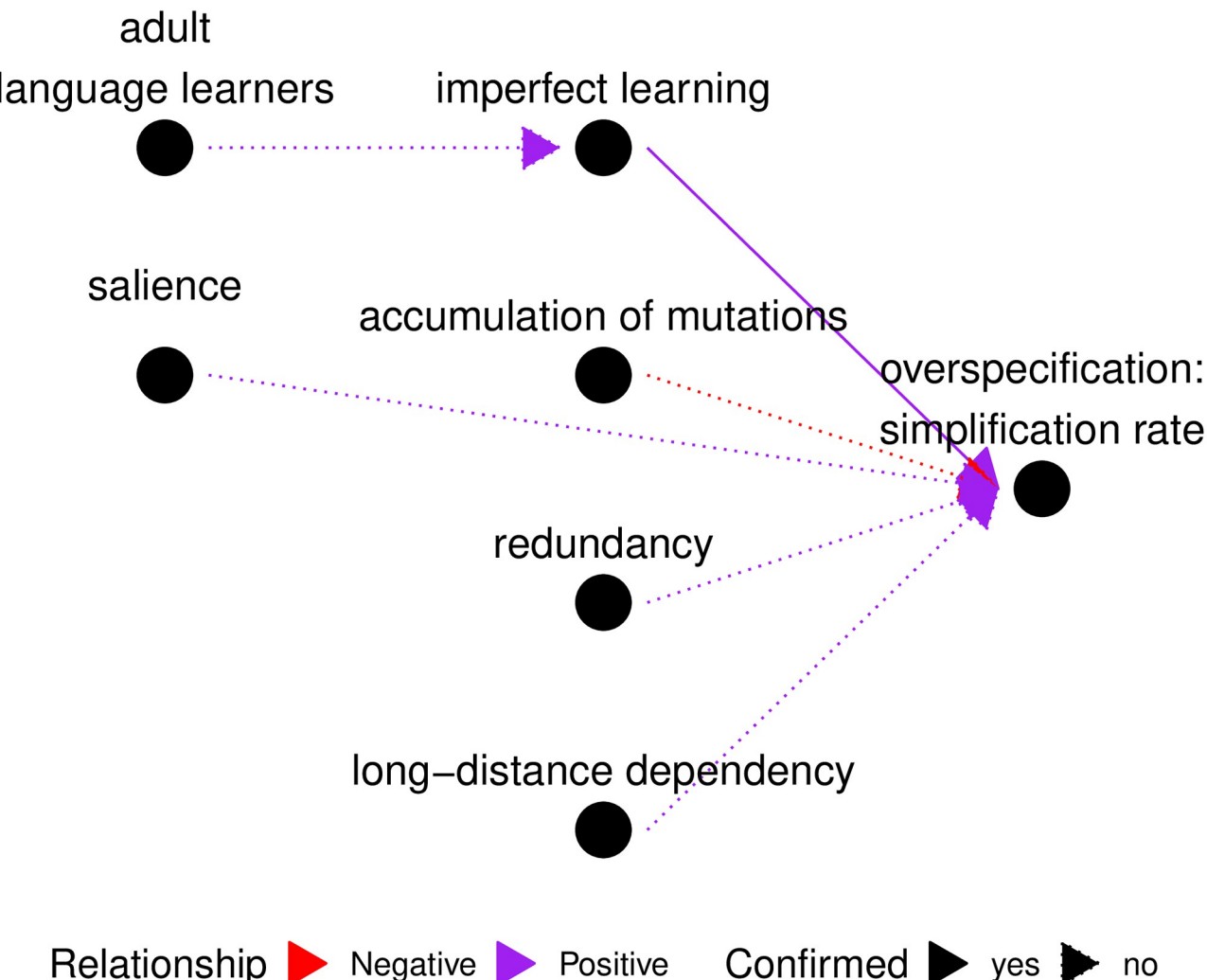

**Fig 8. Causal graph summarizing the paper's main claims.** Nodes represent phenomena or factors, colons are used to indicate higher-level notions. Edges show how the nodes are correlated. Red arrows show negative correlation, while purple arrows show positive correlation. Solid arrows mean that our study provides direct empirical evidence in favour of the correlation, dotted arrows mean that the correlation is hypothetical. We show, for instance, that imperfect learning causes simplification (thus the solid purple arrow). We assume that accumulation of non-systematic mutations may inhibit simplification (thus the dotted red arrow).

contact, affect the loss and maintenance of linguistic complexity. Most, if not all, theories that posit such a causal link assume that imperfect learning is a key factor in the simplification mechanism. Much of the existing evidence, while compelling, is indirect, being based, for instance, on correlational analysis of typological data. Recently, however, a turn towards obtaining more direct insights into causal mechanisms, for instance, through experimental studies, has been observed.

We contribute to this approach, demonstrating by means of a large-scale iterated-learning experiment that imperfect language learning does cause morphological simplification. Complexity decrease was observed in the conditions (T and P) where transmission chains contained generations of imperfect learners (modeled as learners with reduced learning time), but not in the condition where such generations are absent.

The decrease was gradual and partial, i.e. complexity did not get fully eliminated. The decrease mostly affected double agent-marking on verbs, probably because of a unique combination of properties: it is redundant, more difficult to learn due to a long-distance dependency and less salient due to its frequency, length and position. No decrease in complexity was observed for nouns.

In our setting, there were no significant differences between the two interrupted conditions, one of which had three generations of imperfect learners and the other had nine.

To sum up, we show that imperfect learning can be one of the mechanisms that affect changes in morphological complexity, and thus the distribution of complexity across the world's languages. Our results and theoretical considerations suggest, however, that it is unlikely to be the only mechanism and thus its interactions with other potential factors should be further investigated.

In the broader context, our paper presents experimental evidence providing further support to the claim that extra-linguistic factors shape linguistic structure.

## Supporting information

**S1 Appendix. Dataset; detailed results; code.**
(ZIP)

**S1 Text. Instructions to the participants.**
(DOCX)

**S2 Text. Recruiting and filtering participants in an online setting.**
(DOCX)

**S3 Text. The structure of the input languages.**
(DOCX)

**S4 Text. Comprehension test.**
(DOCX)

**S5 Text. Comprehension rate and underspecification rate.**
(DOCX)

**S6 Text. Further discussion of model validity.**
(DOCX)

**S7 Text. Type-token ratio.**
(DOCX)

**S8 Text. Entropy.**
(DOCX)

**S9 Text. Finer-grained analysis of how meanings are expressed: Expressibility.**
(DOCX)

**S1 Fig. Results of the comprehension test.**
(DOCX)

**S2 Fig. Change in underspecification over time.**
(DOCX)

**S3 Fig. Change in transmission fidelity over time.**
(DOCX)

**S4 Fig. Change in entropy over time.**
(DOCX)

**S5 Fig. Change of expressibility of the four categories over time.**
(DOCX)

## Acknowledgments

We are grateful to the popular-science portal "Elementy" and its editor-in-chief Elena Marty-nova for advertising the experiment, to Tanja Russita for designing the Epsilon fauna, to Laura Janda, Hanne Eckhoff, Marc Tang, Natalia Mitrofanova, Seana Coulson, Josefin Lindgren, Tore Nesset, Niklas Edenmyr, Anastasia Makarova, Harald Hammarström, Julia Kuznetsova, and the UCSD Cognation lab members for discussing earlier versions of the article (all remaining errors are ours), and to all beta-testers for helping with pilot studies. We would also like to thank the editor, Vera Kempe, and three anonymous reviewers.

## Author Contributions

**Conceptualization:** Aleksandrs Berdicevskis, Arturs Semenuks.

**Data curation:** Aleksandrs Berdicevskis, Arturs Semenuks.

**Formal analysis:** Aleksandrs Berdicevskis, Arturs Semenuks.

**Funding acquisition:** Aleksandrs Berdicevskis.

**Investigation:** Aleksandrs Berdicevskis, Arturs Semenuks.

**Methodology:** Aleksandrs Berdicevskis, Arturs Semenuks.

**Project administration:** Aleksandrs Berdicevskis.

**Software:** Arturs Semenuks.

**Visualization:** Arturs Semenuks.

**Writing – original draft:** Aleksandrs Berdicevskis, Arturs Semenuks.

**Writing – review & editing:** Aleksandrs Berdicevskis, Arturs Semenuks.

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
