## [Decision Letter · Decision Letter 0]

22 Sep 2021

PONE-D-21-26759Imperfect language learning reduces morphological overspecification: Experimental evidencePLOS ONE

Dear Dr. Berdicevskis,

I have received three reviews who are all very complimentary of your submission and recommend publication. The reviewers are especially impressed with the number of generations and chains, and I concur that this inspires confidence in your data. The reviews differ with respect to how many improvements they recommend for your paper. I therefore am sending it back in the hope that you can address the reviewers’ concerns. In particular, I urge you to consider the following:

1.Both R1 and R2 agree that ‘imperfect learning’ and how it was operationalised need to be defined more clearly early on, in the Abstract and the Introduction. It would also be important to be clearer about whether this is a realistic operationalisation of L2-learners’ deficits and how it links with language contact situations, as R2 notes.

2.Consider more carefully potential confounds between nouns and verbs which could impact comparability of complexity reduction, as R3 notes, and comment on these explicitly.

There are a few more questions that the reviewers raised that I also hope you will be able to address.

We look forward to receiving your revised manuscript.

Kind regards,

Vera Kempe

Academic Editor

PLOS ONE

Journal Requirements:

4. We note that you have referenced (ie. Meinhardt, E., et al.) which has currently not yet been accepted for publication. Please remove this from your References and amend this to state in the body of your manuscript: (ie “Meinhardt, E. et al. [Unpublished]”) as detailed online in our guide for authors

Reviewers' comments:

Reviewer's Responses to Questions

**Comments to the Author**

1. Is the manuscript technically sound, and do the data support the conclusions?

Reviewer #1: Yes

Reviewer #2: Yes

Reviewer #3: Partly

2. Has the statistical analysis been performed appropriately and rigorously? 

Reviewer #1: Yes

Reviewer #2: Yes

Reviewer #3: Yes

3. Have the authors made all data underlying the findings in their manuscript fully available?

Reviewer #1: Yes

Reviewer #2: Yes

Reviewer #3: Yes

4. Is the manuscript presented in an intelligible fashion and written in standard English?

Reviewer #1: Yes

Reviewer #2: No

Reviewer #3: Yes

5. Review Comments to the Author

Reviewer #1: This paper presents an exciting iterated language learning study testing the hypothesis that imperfect learning reduces the morphosyntantic complexity (specifically, overspecification) of languages. The authors manipulate the degree of “interrupted transmission” in transmission chains seeded with perfectly regular (moreover, overspecified) languages and find support for their main hypothesis.

This is a well-conducted study that tests an interesting hypothesis that is often mentioned in the literature but has not been directly experimentally tested. The introduction presents a rich, relevant and up to date literature review. The design, materials and analysis are sound and clearly explained, I commend the large number of chains tested. The claims made are warranted by the results; the SM provide a wealth of additional detail and the paper is carefully and clearly written. I think this paper will be of great interest to the language evolution nad cultural evolution communities and therefore I recommend acceptance for publication.

I have a couple of very minor comments:

In the abstract, make it explicit that you experimentally manipulate imperfect learning, otherwise it might be understood that you measured this variable post-hoc (e.g. some participants learned less perfectly than others, and you hypothesise that those will reduce complexity.) E.g., “…next generation. We manipulate the learning time of learners and show that when transmission chains …”

Typo:

p. 18 para 4, line 1 “the fact that a small decrease”

Reviewer #2: The paper presents an interesting artificial language study addressing a possible mechanism that could drive the simplification of morphological systems in natural languages.

They present evidence from an iterated learning experiment that shows that transmission chains including imperfect learners show a faster decrease in complex features, specifically overspecified markers.

I am generally very sympathetic towards the paper. The question is relevant with respect to ongoing discussions about which factors shape linguistic structure and there are still few experiments specifically trying to isolate aspects of simplification and complexification. The number of chains is impressive for an iterated learning study, and the results are interesting, since they support the idea that small changes introduced by learners gradually accumulate over cultural transmission. I also liked the fact that three conditions were chosen that nicely illustrate that the observed simplification is a gradual phenomenon that could relate to small statistical changes that accumulate and lead to variation specifically in scenarios like language contact.

However, I think the paper could be improved in various ways. Firstly, I think the introduction could provide a clear definition of ‘imperfect learning’ early on, to better frame the experiment. Although it becomes clear later when the methods are described, it would have been nice if the term was defined early on to make the main hypothesis very explicit and better understand what the design was supposed to model.

I also would like to see a more detailed discussion of what the author’s think this looks like in the real world, i.e. how the type of imperfect learning observed here fits into actual language contact and change, and how it would interact with some of the other factors they discuss. I also wonder how the experiment was affected if interaction was introduced at every generation of the chains. Kirby et al. 2015 and Motamedi et al 2019 have shown that both communication and transmission are important to get systematic, efficient, and structured languages. There is an element of pseudo-communication in the experiment (as described in the supplementary materials), but I wonder to what extent real communication would alter the results, and some discussion on the roles of interaction and transmission in real-world scenarios where imperfect learning is present.

Furthermore, I find the claim of the ‘weak trend’ of complexification for nouns a bit problematic given the statistical results. It would be more appropriate to explicitly state that while there appears to be no reduction, and numbers point even into the opposite direction, the results do not support a complexification effect, and from visual inspection of Fig, 6 it also does not look like such an effect would be present. However, the positive slope could inspire a replication experiment to establish whether it is robust. To this end, a power analysis could be conducted to determine the necessary N to detect such a weak trend if it is there. One could also consider specific experiments designed to address this difference between nouns and verbs, and I would appreciate if the paper included more of an outlook discussing how the present findings could be integrated with existing literature and followed up. The authors mention CHIELD, where several causal graphs on morphological complexity are included. It would be nice to see a discussion of how their graph fits in there. Said causal graph should also be described/paraphrased briefly in the text body to make it easier to understand.

Lastly, I wonder to what extent it mattered that the experiment was online. The authors report pilots. Where these also online? If not, I’d be curious what they think the difference was and whether this would mean that online data collection is problematic for artificial language learning experiments. I also wonder whether the fact that participants were forbidden to take notes isn’t in conflict with the fact that their performance on the learning task would increase their odds at winning the lottery, which would create an incentive to cheat (hard to check online).

Regarding the measure of complexity, the authors describe on p. 10 that they use Type-Token Ratio due to Bentz et al.’s recommendation, but I wonder why they haven’t considered using entropy as a second measure to compare the results (which could also be informative when testing for parts-of-speech differences).

Overall, I think this is an interesting study, and I would recommend this with minor revisions and happily accept the paper if the authors streamline introduction and discussion a bit and clarify the points I addressed in this review.

Minor points/typos:

I highly recommend proofreading the entire manuscript and supplementary files, since I probably didn’t catch all mistakes.

p.2 ‘[other] than in its on terms’

p.3 remove redundant parethesis for Roberts & Winters 2013

p.3 ‘Through this, we test whether imperfect learning can contribute to the explanation of the distribution of morphological complexity across the languages of the world.’

Semantics: It’s not imperfect learning that contributes to the explanation, but our understanding of it. (Fix: … we test whether imperfect learning shapes the distribution of …)

p.5 ‘It is limited to stems (names for entities and events) and number marking, the only instance of overspecification in our languages is the redundant agent-marking on verbs.’ These should be two sentences.

p.5 ‘to what extent [does] the degree of overall simplification depend(s)…

p.5 ‘we go online’ ‘ we recruited’ Tense is not used consistently throughout the article. Generally I would advise to keep the *discussion* and theoretical point in present tense, and the *reporting*, i.e. description of the experiment conducted and analysis performed in past tense, as is usually the convention for experimental reports.

p. 7 ‘real human languages’ -> natural languages

p. 10 ‘[their] learner type is confounded with the complexity’

p. 14 ‘For nouns in T condition, there is …’ -> make it either ‘in [the] T condition’, or make it consistent with the table, ‘nouns in condition T’

p. 14 ‘The slope is less steep in N condition.’ Same here (and also some other parts of the paper below)

p. 14 Table 3 doesn’t include italics or am I mistaken? Therefore, the caption confused me and I would remove the part about italics.

Reviewer #3: This study is set to test whether imperfect learning leads to morphological simplification. The authors test this hypothesis using iterated learning paradigm with artificial language learning task. They manipulate learning (causing imperfect learning) by exposing number or all generations in the chain to less repetitions of utterances. Morphological complexity is measured as type token ratio (TTR). A second hypothesis the authors test is whether morphological simplification occurs more in the redundant marking in the language (agent marking on verbs). They conclude that morphological simplification occurs more in chains including imperfect learning and that simplification is caused mainly by eliminating redundant agent marking on verbs in the produced languages.

Regarding the way Hypothesis 1 (imperfect learning leads to simplification) is tested I have only minor concerns for the authors to consider whereas I have more major methodological worries regarding Hypothesis 2 and the way it is tested in this paper.

In testing Hypothesis 1 the authors expand on results from Atkinson et al (2018), which they cite, who show that with more exposure to the language learners are able to preserve the complexity of the language, suggesting that morphological complexity is reduced when exposure is limited. In this study, the authors use iterated learning paradigm showing similar results when the transmission chain consists partly or entirely of learners with limited exposure to the language. This is a nice proof of concept of the hypothesis that transmission involving imperfect learners results in simplified languages.

However, a problem that Atkinson et al (2018) attempted to resolve is the problem of linkage, or the mechanism through which the simplified languages produced by individuals affect the languages at the level of the population. Do the authors suggest the iterated learning paradigm as a mechanism to solve the linkage problem? The contribution of paper to the literature beyond results from Atkinson et al would be greater if they explicitly discuss the mechanism they suggest.

The link between social factors discussed in the introduction to the paper and imperfect learning as tested in the experiment should be more clearly described.

I thought that TTR is a good choice as a measure of morphological complexity, although it should be more clearly motivated in the text.

In testing Hypothesis 2, comparing the reduction of marking on verbs and on nouns as a mean of simplification, there are number of differences between verbs and nouns in the way the artificial languages are design in the study. These differences were not accounted for and can also serve as a possible explanation of the results.

First, the number of different nouns in the language (2: round animal and squared animal) is smaller than number of verbs (3). Second, nouns always appear first and before the verb, and third, while number marking on the noun is marked with a consonant, agent marking on the verb is done with a vowel. Altogether, these differences could make the nouns in the language and their marking more salient for learning than the verbs. In this case, it does not have to be related to eliminating redundancy in the language as suggested by the authors, but eliminating parts of the language that were harder to learn and happen to be the redundant marker in this experimental design.

Figure 6 in the paper illustrates the different number of noun vs. verbs in the language and how it affect the initial TTR of the two elements when looked at separately. While The initial (generation 0) TTR value of nouns is less than 0.3, the initial TTR value of verbs is 0.5. This (according to the measure of morphological complexity proposed by the authors) suggests for a difference in the morphological complexity of nouns vs. verbs in the initial language which makes it difficult to compare the two. Therefore, I find it hard to deduce from results shown in this part to general conclusions regarding simplification through elimination of redundancy in the language.

6. PLOS authors have the option to publish the peer review history of their article (what does this mean?). If published, this will include your full peer review and any attached files.

Reviewer #1: No

Reviewer #2: No

Reviewer #3: No

---

## [Author Response · Author response to Decision Letter 0]

8 Nov 2021

We would like to thank the editor and the three anonymous reviewers for the useful comments and suggestions. Please our point-by-point responses below.

EDITOR:

>> 1.Both R1 and R2 agree that ‘imperfect learning’ and how it was operationalised need to be defined more clearly early on, in the Abstract and the Introduction. It would also be important to be clearer about whether this is a realistic operationalisation of L2-learners’ deficits and how it links with language contact situations, as R2 notes.

Response: We added the explanations of how we operationalize imperfect learning to the Abstract and the Introduction. We also added Section 2.7, where we discuss the validity of our model and its relation to the real world.

>> 2.Consider more carefully potential confounds between nouns and verbs which could impact comparability of complexity reduction, as R3 notes, and comment on these explicitly.

Response: We added an explicit discussion of alternative explanations in Section 4.2. We also mitigated the claim that the partial elimination of the double agent-marking is caused by its redundancy and weakened the wording in Section 1.3, framing the former Prediction 2 (now Exploratory question 1) as exploratory and not confirmatory.

REVIEWER 1:

>> In the abstract, make it explicit that you experimentally manipulate imperfect learning, otherwise it might be understood that you measured this variable post-hoc (e.g. some participants learned less perfectly than others, and you hypothesise that those will reduce complexity.) E.g., “…next generation. We manipulate the learning time of learners and show that when transmission chains …”

Response: We changed the wording in the abstract to: "...next generation. Manipulating the learning time showed that when transmission chains..."

>> Typo: p. 18 para 4, line 1 “the fact that a small decrease”

Response: Corrected.

REVIEWER 2:

>> Firstly, I think the introduction could provide a clear definition of ‘imperfect learning’ early on, to better frame the experiment. Although it becomes clear later when the methods are described, it would have been nice if the term was defined early on to make the main hypothesis very explicit and better understand what the design was supposed to model.

Response: We added the definition at the end of Section 1.2. We also added Section 2.6 where we discuss our operationalization of imperfect learning in even more details.

>> I also would like to see a more detailed discussion of what the author’s think this looks like in the real world, i.e. how the type of imperfect learning observed here fits into actual language contact and change, and how it would interact with some of the other factors they discuss. I also wonder how the experiment was affected if interaction was introduced at every generation of the chains. Kirby et al. 2015 and Motamedi et al 2019 have shown that both communication and transmission are important to get systematic, efficient, and structured languages. There is an element of pseudo-communication in the experiment (as described in the supplementary materials), but I wonder to what extent real communication would alter the results, and some discussion on the roles of interaction and transmission in real-world scenarios where imperfect learning is present.

Response: We added Section 2.7, where we discuss these and other limitations and the validity of our model, and its relation to the real world. We also discuss potential relevance of the experiment to the real-world processes in Section 4.4

>> Furthermore, I find the claim of the ‘weak trend’ of complexification for nouns a bit problematic given the statistical results. It would be more appropriate to explicitly state that while there appears to be no reduction, and numbers point even into the opposite direction, the results do not support a complexification effect, and from visual inspection of Fig, 6 it also does not look like such an effect would be present.

Response: We agree that the claim was stronger than the data warranted. We changed the wording approximately as the reviewer recommends. In section 4.2, we still hypothesize which conclusions could have been drawn if the complexification trend was confirmed, but we say explicitly that it was not.

>> However, the positive slope could inspire a replication experiment to establish whether it is robust. To this end, a power analysis could be conducted to determine the necessary N to detect such a weak trend if it is there. One could also consider specific experiments designed to address this difference between nouns and verbs

Response: While we agree that such an experiment would be useful and interesting, we think it should be part of a separate study. We think that describing the design of a potential experiment (and conducting power analysis) in this article would not make our claims clearer. Instead, it will probably look like a detour for the reader (and will also make the article substantially longer).

>> I would appreciate if the paper included more of an outlook discussing how the present findings could be integrated with existing literature and followed up.

Response: We tried to do that in Sections 4.4 and 5.

>> The authors mention CHIELD, where several causal graphs on morphological complexity are included. It would be nice to see a discussion of how their graph fits in there. 

Response: We tried doing that, but the discussion largely repeats what already has been said in Introduction and Discussion. It is possible to add more graphs from the CHIELD and discuss potential usages of the database, but we think that is beyond the scope of this article.

>> Said causal graph should also be described/paraphrased briefly in the text body to make it easier to understand.

Response: We added the description to the caption of Figure 8.

>> Lastly, I wonder to what extent it mattered that the experiment was online. The authors report pilots. Where these also online? If not, I’d be curious what they think the difference was and whether this would mean that online data collection is problematic for artificial language learning experiments. I also wonder whether the fact that participants were forbidden to take notes isn’t in conflict with the fact that their performance on the learning task would increase their odds at winning the lottery, which would create an incentive to cheat (hard to check online).

Response: We agree that running the experiment online results in a range of potential problems, including those mentioned by the reviewer. We discuss these matters in the SI. We have now added an explicit reference to the relevant section of the SI at the end of Section 2.1. Pilot experiments were run both offline and online, but in the offline version, we used a slightly different design (more complex language) and collected just a few datapoints, so we cannot make any meaningful comparisons. In general, the purpose of the pilots was to provide us with intuitive understanding of whether the experiment is implementable (input languages not too difficult and too easy to learn; instructions clear; learning time appropriate etc.). The pilots were not rigorous enough to warrant any formal hypothesis testing, which is why we do not report the data and do not discuss them in the article.

>> Regarding the measure of complexity, the authors describe on p. 10 that they use Type-Token Ratio due to Bentz et al.’s recommendation, but I wonder why they haven’t considered using entropy as a second measure to compare the results (which could also be informative when testing for parts-of-speech differences).

Response: We have now added Text S9 and Figure S4 (and a reference to them from the main text) that describe what the results are if entropy is used as a measure. The main patterns seem to be approximately the same.

>> I highly recommend proofreading the entire manuscript and supplementary files, since I probably didn’t catch all mistakes.

Response: done!

>> p.5 ‘we go online’ ‘ we recruited’ Tense is not used consistently throughout the article. Generally I would advise to keep the *discussion* and theoretical point in present tense, and the *reporting*, i.e. description of the experiment conducted and analysis performed in past tense, as is usually the convention for experimental reports.

Response: The usage of tense harmonized as the reviewer suggests.

>> p. 14 Table 3 doesn’t include italics or am I mistaken? Therefore, the caption confused me and I would remove the part about italics.

Response: The part about italics removed from the caption

>> [Other typos and language-editing suggestions]

Response: All corrected

REVIEWER 3:

>> However, a problem that Atkinson et al (2018) attempted to resolve is the problem of linkage, or the mechanism through which the simplified languages produced by individuals affect the languages at the level of the population. Do the authors suggest the iterated learning paradigm as a mechanism to solve the linkage problem? The contribution of paper to the literature beyond results from Atkinson et al would be greater if they explicitly discuss the mechanism they suggest.

Response: We added an explicit note at the end of Section 4.1 that we are not trying to address the problem of linkage in this study. In general, iterated learning paradigm can of course be used as a means to address it, but that would require more complex experiment design, including at least intra-generational communication. We added discussion of how our paper contributes to the existing literature to Section 4.4.

>> The link between social factors discussed in the introduction to the paper and imperfect learning as tested in the experiment should be more clearly described.

Response: We rewrote Sections 1.1 and 1.2, trying to make it clearer that many (if not all) social factors listed in 1.1 are assumed to facilitate imperfect learning, which is assumed to facilitate simplification (which is what we test). It is possible to shorten discussion in 1.1, focusing, for instance, solely on contact as the social factor, but we would like to provide the reader with a broader overview of which factors are being considered in more theoretical discussions and then narrow down to those which are most relevant for the current study.

>> I thought that TTR is a good choice as a measure of morphological complexity, although it should be more clearly motivated in the text.

Response: We agree that the choice of the measure should be motivated. We added even more details to our explanation of why we chose TTR. Note that we also ran an additional measurement using entropy as a secondary measure, as R2 suggested. The end of Section 2.8 now reads: "Given that there are different means of measuring complexity, each with its own advantages and drawbacks (Berdicevskis et al., 2018), we would like to motivate our choice of measure. TTR is a simple, easily interpretable and reproducible measure, which does not require elaborate theoretical assumptions. It is usually applied to corpora, but given the nature of our artificial languages, it is an adequate measure of their lexicogrammatical diversity. First, by design, each language is a complete enumeration of all possible meanings, i.e. can be construed as a corpus. Second, the distribution of meanings in the Epsilon universe is always uniform, i.e. we do not have to worry about the potential effect of frequency distributions influencing the measure. Finally, TTR is highly sensitive to text size, but since all our languages share the same meaning space, they can be treated as parallel corpora, which resolves the problem. Simplification should then result in the loss of overspecification, i.e. lower TTR. Bentz et al. (2015) describe other measures of lexicogrammatical diversity (Shannon entropy and Zipf-Mandelbrot’s law parameters), but mention that TTR is the most responsive of these three, which is important given the small size of our ”corpora”. We make an additional measurement using entropy, which yields similar results (see Text S9; Fig S4)."

>> In testing Hypothesis 2, comparing the reduction of marking on verbs and on nouns as a mean of simplification, there are number of differences between verbs and nouns in the way the artificial languages are design in the study. These differences were not accounted for and can also serve as a possible explanation of the results. First, the number of different nouns in the language (2: round animal and squared animal) is smaller than number of verbs (3). Second, nouns always appear first and before the verb, and third, while number marking on the noun is marked with a consonant, agent marking on the verb is done with a vowel. Altogether, these differences could make the nouns in the language and their marking more salient for learning than the verbs. In this case, it does not have to be related to eliminating redundancy in the language as suggested by the authors, but eliminating parts of the language that were harder to learn and happen to be the redundant marker in this experimental design. Figure 6 in the paper illustrates the different number of noun vs. verbs in the language and how it affect the initial TTR of the two elements when looked at separately. While The initial (generation 0) TTR value of nouns is less than 0.3, the initial TTR value of verbs is 0.5. This (according to the measure of morphological complexity proposed by the authors) suggests for a difference in the morphological complexity of nouns vs. verbs in the initial language which makes it difficult to compare the two. Therefore, I find it hard to deduce from results shown in this part to general conclusions regarding simplification through elimination of redundancy in the language.

Response: We mitigated the claim that the partial elimination of the double agent-marking on verbs is caused by redundancy. We also weakened the wording in the introduction, framing Hypothesis 2 as exploratory and not confirmatory. In Section 4.2, we review the potential factors listed by the reviewer (and some others).

---

## [Decision Letter · Decision Letter 1]

7 Dec 2021

PONE-D-21-26759R1Imperfect language learning reduces morphological overspecification: Experimental evidencePLOS ONE

Dear Dr. Berdicevskis,

Two of the original reviewers were able to check your revision and are largely satisfied with how you addressed their concerns. At this point I am returning your manuscript for one further very minor review as I would like to urge you to make the following changes in your final version: As originally mentioned by Reviewer 2, and as now reiterated by Reviewer 3, the discussion of increased complexity for nouns is not warranted given the non-significant result so I suggest to defer this to future studies should they show more robust findings in this regard and leave it out of the present submission. In addition, please address the very minor changes suggested by Reviewer 3. Finally, I noticed that Friederici was misspelled in S7.

Once you have made these minor revisions I am hopeful the paper will be ready for acceptance.

We look forward to receiving your revised manuscript.

Kind regards,

Vera Kempe

Academic Editor

PLOS ONE

Journal Requirements:

Reviewers' comments:

Reviewer's Responses to Questions

**Comments to the Author**

1. If the authors have adequately addressed your comments raised in a previous round of review and you feel that this manuscript is now acceptable for publication, you may indicate that here to bypass the “Comments to the Author” section, enter your conflict of interest statement in the “Confidential to Editor” section, and submit your "Accept" recommendation.

Reviewer #1: All comments have been addressed

Reviewer #3: (No Response)

2. Is the manuscript technically sound, and do the data support the conclusions?

Reviewer #1: Yes

Reviewer #3: Yes

3. Has the statistical analysis been performed appropriately and rigorously? 

Reviewer #1: Yes

Reviewer #3: Yes

4. Have the authors made all data underlying the findings in their manuscript fully available?

Reviewer #1: Yes

Reviewer #3: Yes

5. Is the manuscript presented in an intelligible fashion and written in standard English?

Reviewer #1: Yes

Reviewer #3: Yes

6. Review Comments to the Author

Reviewer #1: All the comments and issues raised by the reviewers have been satisfactorily addressed in the revision.

Reviewer #3: The authors have addressed the concerns I raised in the previous round.

There are some minor changes that I suggest at this point -

line 347 - "we actually see a small increase in complexity" make it clear that this was not significant according to your analysis, otherwise this could be misleading, given the results afterward.

line 484 - "We hypothesize that there are two main reasons for that" should be three possible explanations rather than two reasons.

lines 505 to 508 - I don’t think you can draw conclusions from a non significant result, and for that matter speculating on how this non significant observation fits with hypotheses in the literature.

7. PLOS authors have the option to publish the peer review history of their article (what does this mean?). If published, this will include your full peer review and any attached files.

Reviewer #1: No

Reviewer #3: No

---

## [Author Response · Author response to Decision Letter 1]

15 Dec 2021

We thank the editor and the reviewer for the valuable comments.

EDITOR

>> As originally mentioned by Reviewer 2, and as now reiterated by Reviewer 3, the discussion of increased complexity for nouns is not warranted given the non-significant result so I suggest to defer this to future studies should they show more robust findings in this regard and leave it out of the present submission

We remove the discussion and keep only the mention of a potential point of interest for the future studies (see also our response to Reviewer 3 below).

>>Finally, I noticed that Friederici was misspelled in S7.

Corrected!

REVIEWER 3

>> line 347 - "we actually see a small increase in complexity" make it clear that this was not significant according to your analysis, otherwise this could be misleading, given the results afterward.

Reworded as "for nouns, no decrease is observed (there is a very small increase, but it is not significant)"

>> line 484 - "We hypothesize that there are two main reasons for that" should be three possible explanations rather than two reasons.

Yes, of course. Reworded as the reviewer suggests.

>> lines 505 to 508 - I don’t think you can draw conclusions from a non significant result, and for that matter speculating on how this non significant observation fits with hypotheses in the literature.

We removed the speculations (lines 508 to 520) and made it clear that we do not claim that we observed a significant effect and do no draw any conclusions. We still think it is a potentially interesting avenue to explore in future studies and mention this. The current wording is:

"Unlike verbs, nouns do not get simplified. If anything, according to the TTR measure, they become slightly more complex (see Section 3.3), but we cannot claim that this effect is robust and reproducible. This observation, however, may deserve to be further tested in future studies in the light of different hypotheses considering complexification in real languages [citations]"

---

## [Editor Report · Decision Letter 2]

7 Jan 2022

Imperfect language learning reduces morphological overspecification: Experimental evidence

PONE-D-21-26759R2

Dear Dr. Berdicevskis,

Happy New Year! Very pleased this interesting paper will now be out. Standard text below:

--Vera

We’re pleased to inform you that your manuscript has been judged scientifically suitable for publication and will be formally accepted for publication once it meets all outstanding technical requirements.

Kind regards,

Vera Kempe

Academic Editor

PLOS ONE
---

## [Editor Report · Acceptance letter]

13 Jan 2022

PONE-D-21-26759R2 

Imperfect language learning reduces morphological overspecification: Experimental evidence 

Dear Dr. Berdicevskis:

I'm pleased to inform you that your manuscript has been deemed suitable for publication in PLOS ONE. Congratulations! Your manuscript is now with our production department. 

Kind regards, 

on behalf of

Prof Vera Kempe 

Academic Editor

PLOS ONE